# Self-Sensing Rubber for Bridge Bearing Monitoring

**DOI:** 10.3390/s23063150

**Published:** 2023-03-15

**Authors:** Alessandra Orfeo, Enrico Tubaldi, Jack McAlorum, Marcus Perry, Hamid Ahmadi, Hazel McDonald

**Affiliations:** 1Department of Civil and Environmental Engineering, University of Strathclyde; Glasgow G1 1XQ, UK; 2Tun Abdul Razak Research Centre-TARRC, Hertford SG13 8NL, UK; 3Transport Scotland, Glasgow G4 0HF, UK

**Keywords:** rubber, carbon black, Printex, smart bearing, resistivity

## Abstract

Elastomeric bearings are widely used in bridges to support the superstructure, to transfer loads to substructures, and to accommodate movements induced by, for example, temperature changes. Bearing mechanical properties affect the bridge’s performance and its response to permanent and variable loadings (e.g., traffic). This paper describes the research carried out at Strathclyde towards the development of smart elastomeric bearings that can be used as a low−cost sensing technology for bridge and/or weigh−in−motion monitoring. An experimental campaign was performed, under laboratory conditions, on various natural rubber (NR) specimens enhanced with different conductive fillers. Each specimen was characterized under loading conditions that replicated in−situ bearings to determine their mechanical and piezoresistive properties. Relatively simple models can be used to describe the relationship between rubber bearing resistivity and deformation changes. Gauge factors (*GF*s) in the range between 2 and 11 are obtained, depending on the compound and the applied loading. Experiments were also carried out to show that the developed model can be used to predict the state of deformation of the bearings under random loadings of different amplitudes that are characteristic of the passage of traffic over a bridge.

## 1. Introduction

Elastomeric bearings [1,2] are one of the most common types of bearings, with many applications in civil engineering, not limited to supporting bridge decks (e.g., [3,4,5,6]). Although elastomeric bearings are durable and require infrequent maintenance, ensuring that they perform as expected is of paramount importance to bridge safety assessment and management [1,2,7,8]. Current approaches for evaluating the condition of elastomeric bearings rely heavily on visual inspections, but these only provide intermittent, surface−level information. Researchers have therefore proposed alternative monitoring schemes. Soleimani et al. [9] proposed three−dimensional digital image correlation coupled with finite element analysis to identify internal defects in bearings, whereas Topkaya and Akbari [10,11] proposed non−destructive test methods to determine the shear modulus of steel reinforced elastomeric bearings. Attempts have also been made to develop “smart” elastomeric bearings, i.e., bearings equipped with internal sensing systems, capable of providing continuous information on bearing reactions and displacements. Agrawal et al. [7] proposed schemes that use combinations of linear variable differential transformers, pressure sensors, accelerometers, thermocouples, and fibre optic sensors. Other schemes include fibre Bragg grating displacement sensors for bearing condition monitoring [12], piezoresistive sensors for bearing stress distribution measurement [13], and polyvinylidene fluoride polymer film−based sensors for vertical reaction measurement [14].

Equipping bearings with additional sensors is a promising but expensive approach. In this article, we explore the feasibility of exploiting the piezoresistive properties of filled rubber to monitor the condition of elastomeric bearings and of structures supported by them, without any additional sensors. While natural rubber is an insulating polymer, the carbon−filled rubber commonly employed in bearing construction exhibits a good electrical conductivity [15], provided that the volume fraction of the filler is higher than the percolation threshold [16,17]. When this is the case, changes in the strain of the rubber result in changes to its resistivity. The idea of this study is that these changes in resistivity can be used to directly establish the strain state of the rubber, thus turning the bearing into a multifunctional, self−sensing, and low−cost system for monitoring bridges and weigh−in−motion applications. With the same principle, it is also possible to detect whether damage/cracking has occurred within the rubber or at the interface between the rubber and the steel laminates, similarly to what was proposed in other applications where conductive fillers were dispersed in an insulating polymer matrix to create a conductive network (see, e.g., [18,19,20,21]).

The piezoresistivity of rubber has already been investigated in previous studies. [15,22,23,24]. Some of the first studies considered carbon black as a filler considering the impacts of the applied strain direction [15] solvent swelling and temperature changes on electro−mechanical response [22] and the response to cyclic loading [23]. Giannone et al. [25] also investigated the possibility of using conductive rubber filled with carbon black to develop piezoresistive bearings, and highlighted the creep behaviour of electrical resistance. The results of these studies showed that bearings containing carbon black fillers used in rubber for engineering applications do not exhibit a reversible strain−sensing response, i.e., there is not a clear monotonic relationship between the change of strain and the change of resistivity, with a different path followed upon loading and unloading. However, a reversible behaviour is essential in order to develop rubber−based strain−measuring devices. Subsequent studies have therefore investigated rubber compounds incorporating alternative carbon black fillers that provide more reversible behaviour. Jha et al. [24] showed that rubber filled with Printex XE2, which is usually used for inks in the printing industry, possesses quite good reversible properties under tensile loading, but they did not characterise the piezoresistive behaviour of the compound under compressive or shear loading.

Therefore, more research is needed to further explore the piezoresistive properties of filled rubber under loading conditions that are more representative of those experienced by laminated bearings in bridges. This paper’s aims are to fill this gap and illustrates the results of the experimental campaign carried out to characterise the piezoresistive properties of specimens made with different rubber compounds under shear and compression loadings in addition to tensile loadings. Section 1 of the paper outlines the various filled rubber compounds tested (made using either carbon black or Printex XE2 as filler) and the experimental setup used to evaluate the piezoresistive properties of the filled rubber specimens. Section 2, Section 3 and Section 4 of the paper describe the results of the electro−mechanical tests carried out on the specimens made with the different compounds.

The work described here could be of great benefit to bridge asset managers, as bridge responses to external loadings, such as those induced by the traffic or the external environment, are influenced by bearing mechanical properties. Thus, the measurements of stresses and strains in bridge bearings can provide useful information not only on the bearing condition, but also on the health of the bridges [1,26,27,28]. It can also constitute an indirect means for estimating the traffic on bridge superstructures (weigh−in−motion) [29,30].

## 2. Specimens and Experimental Setup

This section describes the investigated rubber compounds, the various specimens made with them, and the electro−mechanical testing apparatus used to assess their piezoresistive behaviour.

### 2.1. Compounds

Table 1 shows the formulation of the compounds used in the study, which differ mainly for the type of filler used. The first one (denoted as carbon black 70) is a more traditional compound that is used to develop laminated rubber bearings, whereas the other two are more innovative compounds that use Printex XE2 to enhance the piezoresistive properties of rubber and achieve a more reversible behaviour (i.e., similar electrical response upon mechanical loading and unloading). The quantity of filler added is directly related to the conductivity of the final material. For a given type of filler, it is possible to identify the “percolation threshold”, i.e., the volume fraction of filler which must be exceeded to allow the compound to become an electrical conductor (as shown in Figure 1). In the region of the percolation threshold, the sensitivity of the resistivity to mechanical deformation is highest, whereas it decreases for volume fractions of filler higher than the percolation threshold. The percolation threshold for Printex XE2 is much lower than for carbon black due to the hollowed−out shell−like structure of Printex XE2, which results in a higher surface area and a more dramatic effect in terms of electrical conductivity properties [24,31].

The mixing of the compounds was carried out in a Polylab mixer, at 50 °C temperature and at 60 RPM. Printex XE2 was sonicated in an UltraSonic bath, even though the comparisons of the transmission electron micrographs for rubber with dry (unsonicated) Printex did not show significant differences in terms of dispersion.

The amount of filler in the compound is measured by the parts per hundred of rubber (phr). Jha et al. [24] already investigated the piezoresistive behaviour of Printex XE2−filled rubber tensile specimens, but with a phr of Printex of 10, which is very close to the percolation threshold. The phr values considered in the present study are slightly higher (12 and 15 phr) in order to achieve a more dissipative and stiff behaviour for the compound, as this is typical of laminated bearings used in construction practice. It is noteworthy that high energy dissipation capabilities are desirable for isolation bearings (see, e.g., [5,6]), while they would not be strictly required by bridge bearings in non-seismic zones.

### 2.2. Test Pieces

Three different types of test piece were manufactured and tested under different loading conditions, namely tensile, compressive, and shear, while measuring the changes in electrical resistance. In particular, the changes in resistivity were measured during tensile tests for natural rubber specimens filled with carbon black, for natural rubber specimens filled with Printex XE2 at 15 phr, and during shear and compression tests for the compound with Printex XE2 at 12 phr. Table 2 summarises the compounds and test pieces considered and the loading conditions they have been subjected to.

The experimental set−up for this test is shown in Figure 2. The test specimens considered for the tensile tests are rectangular strips of rubber cross−section, with 80 mm × 25 mm sides and a thickness of 1.1 mm. The electrical resistivity was measured with a four-point contact method.

Double bonded shear (DBS) specimens were used for the double shear tests. They consist of two cylindrical rubber discs moulded between three brass pieces. The thickness of the disc is 6 mm and its diameter *d_0_* is 25 mm (Figure 3a). The compressive specimens consist of one cylindrical rubber disc moulded between two brass pieces and the diameter *d*_0_ is 50 mm (Figure 3b). Two compressive specimens filled with Printex 12 phr have been manufactured with two different rubber layer thickness, 5 mm and 8.47 mm, corresponding to the height over diameter ratio *l*_0_/*d*_0_
*=* 0.1 and *l*_0_/*d*_0_
*=* 0.17, respectively. It is noteworthy that brass was used instead of steel for the end pieces, as it could be bonded directly to the rubber without the need for bonding agent which would have affected the resistivity measurements.

The electrical resistivity of the DBS and compressive specimens was measured with a two−point contact method and the experimental set−up is shown in Figure 4a,b. In this technique, the specimen resistance is measured by two electrodes. A direct current is applied and the drop in the potential between the electrodes is measured. Figure 5 illustrates a schematic representation of the experimental resistance measurement of the DBS and compressive rubber specimens.

Tensile, double shear, and compression experiments were applied in up to five identical cycles to a “virgin” rubber specimen, never tested before. This way, it was possible to evaluate the effect of “stress−softening”. This effect takes place within the untested rubber during the first deformation paths, which are generally characterized by a higher stiffness and dissipative capacity [5]. This effect usually reached equilibrium after a few cycles.

### 2.3. Stress-Strain Response and Resistivity Measurement

Mechanical tests were carried out by imposing displacements on the uniaxial test pieces. The mechanical response is described in terms of nominal stress vs. stretch ratio. The nominal stresses are obtained by dividing the axial force in tension (or compression) *F_p_* by the initial cross−sectional area of the uniaxial (or compressive) specimen *A*_0_:(1)$$\sigma =\frac{F_{p}}{A_{0}}$$

The stretch ratio is defined as follows:(2)$$\lambda =\frac{l}{l_{0}}$$
where *l* is the stretched length.

The shear tests were carried out on the DBS specimens by imposing different nominal shear strain amplitudes *γ*, defined as:(3)$$\gamma =\frac{u_{s}}{h_{0}}$$
where *u_s_* is the shear displacement, and *h*_0_ is the thickness of a single disc.

The correspondent nominal shear stress is then obtained by dividing the transverse reaction force by two times the initial cross−sectional area of each rubber disk:(4)$$\tau =\frac{F_{t}}{2A_{0}}$$

With regards to the resistivity measurement, direct current (dc) of amplitude *I* was imposed during mechanical testing to interrogate the system and the variation of the electrical resistance *R* was measured during the test via changes in measured voltage, *V*:(5)$$R=\frac{V}{I}$$

Another approach for characterizing the electrical properties of the rubber compounds would be by assessing the electrical impedance, which can be achieved by applying an alternating current (ac) potential and then measuring the phase and amplitude ac voltage response current, which carries more information on the system. In particular, the impedance is a combination of the resistance and of the reactance, which measures the opposition of the system to changes in electric current [32].

The electrical resistivity of the rubber is a material property calculated as follows:(6)$$\rho =\frac{RA_{r}}{t_{r}}$$

The value of *A_r_* and *t_r_* to be considered depends on the type of loading. For uniaxial tensile and compression loading, *A_r_* is the actual (i.e., deformed) cross−sectional area:(7)$$A_{r}=A_{0}/\lambda$$
and *t_r_* coincides with the stretched (or compressed) length *l*.

In the case of double shear tests (two identical resistances arranged in series), the area to be considered in Equation (6) is:(8)$$A_{r}=A_{0}=\pi \frac{d_{0}^{2}}{4}$$
where *d*_0_ is the diameter of the rubber layer.

The length *t_r_* corresponds to the total thickness of a rubber discs, i.e., 2*h_0_*.

## 3. Experimental Tests on Rubber Filled with Carbon Black

### 3.1. Full Cyclic Uniaxial Tensile Tests

Two cyclic uniaxial tensile tests were initially conducted on 80 mm long specimens containing carbon black. In both cycles, the sample was extended in 4 mm steps up to the desired elongation of *λ* = 2. A dwell time of 180 s was considered between each further application of stretch. Figure 6a shows the stress-extension relation obtained during loading and unloading for both cycles. The stresses are nominal stresses, defined as the load divided by the cross−sectional area, *A*_0_, and thus not accounting for the change of the transverse area during loading. The plot exhibits significant hysteresis, as expected for such a type of filled rubber. Moreover, stress−softening behaviour is evident, with the value of the peak stress at the second cycle of approximately 80% of the peak stress at the first cycle. Following the unloading path, the sample buckled for *λ* ≈ 1.15, as shown in Figure 6b. This instability can be explained with the viscous behaviour of the material. When the sample is unloaded after being stretched, there is some residual set due to viscous deformations, and since the sample cannot bear any compressive load, it buckles. It is noteworthy that the initial length of the specimen was recovered after a period of rest of few hours.

Figure 7a,b show the electrical resistivity as a function of the extension ratio obtained for the first and second cycles, respectively.

These data show the relationship between the hysteretic mechanical and electrical behaviours of the material. During the 1st cycle, the resistivity starts to rise with the stretching of the untested rubber due to the breakdown of the carbon black network into smaller aggregates. This increase is then followed at higher extensions by a reduction of the resistivity [15]. During unloading, the electrical resistivity increases to values that are higher than those observed during the initial loading. In the second cycle, the stress−softening effect, related to the breakage of the bonds in the untested rubber, is negligible. After one cycle, the piezoresistive behaviour of the rubber is more predictable, in the sense that the resistivity decreases upon loading, and increases upon unloading. This behaviour can be explained by the fact that the carbon black aggregates, which are the conductive component of the compound, tend to align when the sample is stretched, thus decreasing the resistivity [15].

### 3.2. Uniaxial Tensile Tests: Triangular Input

Another test was performed on the same carbon black specimen. This was initially stretched up to *λ* = 1.2, then the stretch was kept constant for a dwell time of 6 min, and finally the stretch pattern shown in Figure 8a, consisting of positive and negative triangular inputs, was applied. Figure 8b shows the obtained stress-stretch curve. The relaxation behaviour of the rubber for *λ* = 1.2 is evident, as is the significant hysteresis.

Figure 9a shows the time histories of the stretch normalised by *λ* = 1.2, and of the stress and electrical resistivity normalised by their correspondent value at *λ* =1.2. Following the initial stretch, there is an overall trend of a reduction of stresses with time, due to the relaxation behaviour of the rubber. Then, the stresses, respectively, increase and decrease for increasing and decreasing stretch levels. There is also a general trend of a reduction of resistivity with time, which is more significant than for the stresses. However, it can be observed that the resistivity increases when both an increase of stretch and a decrease of stretch are applied, starting from *λ* = 1.2. The different behaviour of the stresses and of the resistivity is highlighted in Figure 9b. It is also noteworthy that the sensitivity of the resistivity, defined as the increase of resistivity divided by the change of stretch, is higher when the specimen is shortened, i.e., by passing from *λ* = 1.2 to *λ* = 1.16, than when it is stretched further, i.e., by passing from *λ* = 1.2 to *λ* = 1.24.

Another test was performed on a new specimen of the same carbon black−filled material and geometry. This was stretched up to *λ* = 1.2. The stretch was kept constant for a dwell time until a constant resistance value was obtained. Then, the sample was stretched again up to different values of *λ* (reported in Table 3) by imposing a series of positive triangular displacement inputs rather than positive and negative inputs as in Figure 8a. Table 3 shows the maximum stretch and the strain rate $\dot{\lambda \;}=\frac{\Delta \lambda }{\Delta t}$ applied in each test.

Figure 10 shows the stress−stretch curves corresponding to the various tests. For stretch levels higher than 1.6, the strain−crystallisation effect can be observed, resulting in a significant increase of stresses for increasing *λ*.

Figure 11 illustrates the resistivity against the extension ratio obtained for Test 1 and Test 4. These tests show again that there is not a clear relationship between elongation and change of resistivity. In general, the results of this first experimental campaign confirmed that typical rubber compounds incorporating carbon black as a filler are characterized by a complex behaviour and phenomena, such as stress−softening, relaxation, and hysteresis. All these phenomena are somehow also reflected in the piezoresistive behaviour of the rubber, which is highly nonlinear. Thus, it is not possible to establish a simple univocal relationship between the change of stress/strain and change of resistivity, and for this reason, the first investigated compound is not suitable for the development of smart elastomeric bearings.

## 4. Experimental Tests on Rubber Filled with Printex 15 Phr

### 4.1. Uniaxial Tensile Tests

Five cyclic uniaxial tensile tests were conducted considering the compound filled with Printex (15 phr). In all cycles, the specimen was extended by 8 mm steps up to the desired elongation *λ* = 2. A dwell time of 60 s was considered between each further application of stretch. Figure 12 shows the stress−stretch curve obtained during loading and unloading for both cycles. The compound is softer than the one with carbon black as filler. Nevertheless, it exhibits some hysteretic behaviour. Moreover, some stress−softening is observed, with a stable response attained after three cycles of deformation. The peak stress at the third cycle is approximately 90% of the peak stress at the first cycle. Compared to the carbon black−filled compound, both the hysteretic capacity and stress−softening are lower.

Figure 13a shows the electrical resistivity as a function of the extension ratio during the loading and unloading part of the 5th cycle. Resistivity peak data at each extension ratio are also shown in Figure 13b for the loading and unloading paths of the 5 subsequent cycles. It can be observed that, apart from the 1^st^ cycle, all subsequent cycles are almost indistinguishable. The electrical resistivity increases with the extension ratio upon loading and reverts to the same initial value along a similar path upon unloading. As observed by Jha et al. [24], this behaviour is explained by the fact that the filler network is not permanently altered under strain. This is different from the case of carbon black−filled rubber, where the strain breaks down the agglomerate structure in the rubber, leading to a net reduction in the number of conduction paths made through the sample.

Figure 14 shows the variation of the resistivity normalized by the resistivity at zero stretch *ρ*_0_ = 0.14 Ωm, i.e., $\Delta \rho /\rho_{0}$ vs. the strain *λ*. The plotted data refer to the tests of Figure 13a. An interpolating function is fitted to the observed data with the aim of developing a model that relates the changes of resistivity to the changes of stretch. The function has a linear trend of variation:(9)$$\frac{\Delta \rho }{\rho_{0}}=9.27\lambda -9.27$$

In order to characterize the sensitivity of the Printex−filled specimen, the gauge factor (*GF*) is evaluated, which is defined as:(10)$$GF\;=\frac{\Delta \rho_{0}/\rho_{0}}{\varepsilon }$$
where *ε* = 1 − *λ* represents the applied strain. The value of the *GF*, according to Figure 14, is equal to 9.3.

Furthermore, the sensing precision is evaluated by evaluating the standard deviation of the residuals in terms of $\Delta \rho /\rho_{0}$, and then dividing it by the *GF*, as shown in Equation (11).
(11)$$\sigma_{\varepsilon }=\frac{\sigma_{\Delta \rho /\rho_{0}}}{GF}$$

The obtained value of $\sigma_{\varepsilon }$ is 0.16 and measures the uncertainty in the estimate of ε given the resistivity measurement.

### 4.2. Uniaxial Tensile Test: Random Input

A second test was performed on the same Printex−filled specimen by imposing the input shown in Figure 15a. This history was designed to be representative of the effect of the traffic passing over a bridge superimposed onto permanent loading. Various stretch ratios are applied to simulate vehicles of different mass. Figure 15b shows the stress and electrical resistance behaviour due to the applied displacement pattern. It can be observed that the resistivity change follows the same pattern as the stress and stretch change.

The purpose of this test is to understand whether by measuring the changes in the electrical resistivity of the specimen if it is possible to reconstruct the strain history it was subjected to, or at least to infer the maximum strain amplitudes it experienced. This is an essential requisite for using the rubber compound to develop smart rubber bearings. Figure 15a also compares the imposed stretch history and the corresponding history inferred by measuring the electrical resistivity changes of the specimen and using Equation (9) to relate these changes of resistivity to changes of stretch. The predicted changes of deformation are in good agreement with the experimental values, thus confirming the promising potential of the compound for the development of self−sensing bearings.

## 5. Experimental Tests on Rubber Filled with Printex 12 Phr

This section describes the tests carried out on double shear and compressive specimens made with Printex with 12 phr. This compound incorporates fewer fillers than the previous one in order to improve the piezoresistive behaviour and the sensitivity of the resistivity to the strain. Nevertheless, the fraction of the filler is higher than the one considered in Jha et al. [24], which was found to be higher than the one corresponding to the percolation threshold.

### 5.1. Double Bonded Shear (DBS) Tests

The DBS tests consisted of 6 cycles carried out for increasing levels of the maximum nominal shear strain amplitude *γ_max_*, namely 0.05, 0.1, 0.2, 0.5, 0.7, and 1. A dwell time of 180 s was considered between each further application of shear strain to allow the rubber to relax.

After the test with *γ_max_* = 1, the tests were performed again to evaluate the softening effect (often referred to as scragging [33]) of exposing the sample to large shear strain amplitudes on the mechanical behaviour and resistive behaviour at strains below the scragging strain of *γ_max_* = 1. Figure 16 shows the nominal shear stress−strain relation obtained during loading and unloading for the 6 cycles at different amplitudes, before and after the test at *γ_max_* = 1. It can be observed that the stress−softening is quite significant at low strain amplitudes (Figure 16a) and negligible at higher amplitudes (Figure 16b).

Figure 17 shows the electrical resistivity as a function of the shear strain during the loading and unloading part of the 6 cycles at *γ_max_* = 0.2, and *γ_max_* = 1 applied after pre−scragging the rubber. In general, the resistivity increases with the shear strain and exhibits an almost reversible behaviour, in agreement with the tests carried out by Jha et al. [24] on tensile specimens and with the tensile tests carried out on the rubber compound with 15 phr of Printex. Only for low strain amplitudes are the changes of resistivity low, which indicates a low sensitivity of the compound (low gauge factor). Nevertheless, such low strain levels are not of interest in bearing applications, since the usual values of the maximum design shear strain are of the order of 50% (see, e.g., [34]), while they are usually higher for seismic isolation bearings [4].

Figure 18 shows the variation of the resistance normalized by the resistivity at zero strain *ρ*_0_ = 7.2 Ωm, i.e., $\Delta \rho /\rho_{0}$, vs. the shear strain *γ*. The plotted data refer to the test of Figure 17b (*γ_max_* = 1). The data are interpolated using a linear functions valid for *γ* in the range between 0 and 0.7, and a quadratic function for *γ* in the range between 0.7 and 1, with the aim of developing a model for relating the changes of resistivity to the changes of shear strain.

Another test was performed on the same DBS specimen by imposing the input of Figure 19a. Figure 19b shows the stress and electrical resistance behaviour due to the applied displacement pattern.

The regression model developed based on the results of the previous tests (shown in Figure 18) is used to infer the changes of strain based on the readings of the resistivity. Figure 20a shows the measured changes of resistivity vs. the imposed changes of strain. In the same figure, the curve corresponding to the model of Figure 18 is shown. Figure 20b shows the time history of the experimental and predicted shear strains, which are in reasonable accord with each other. It is noteworthy that the proposed model cannot predict good accuracy cycles with low deformation amplitudes due to the low sensitivity of $\Delta \rho /\rho_{0}\;$ to shear strains for small *γ* values.

### 5.2. Compression Tests on Rubber Filled with Printex 12 Phr (Specimen with l_0_/d_0_ = 0.1)

Cyclic compressive tests were conducted on the cylindrical specimen with a relatively thick rubber layer (*l*_0_ = 5 mm), corresponding to an aspect ratio of *l*_0_*/d*_0_ = 5/50 = 0.1. The sample was compressed by 0.5 mm steps (Δ*λ* = 0.166) up to the desired compression ratio *λ* = 0.6. A dwell time of 60 s was considered between each further application of compression. Three identical cycles were imposed. Figure 21 shows the stress–compression ratio curve obtained for all the cycles. Some minor hysteresis and stress−softening behaviour is observed, with the peak stress at the second cycle equal to approximately 96% of the peak stress at the first cycle.

Figure 22 shows the electrical resistivity as a function of the compression ratio during the first 3 cycles. The relation between the two quantities is highly nonlinear, with small changes of resistivity for values of *λ* higher than 0.8. However, a clear trend can be identified, and the behaviour is overall quite reversible.

Another test was performed on the same sample. It was compressed up to *λ* = 0.97, then the compression was kept constant for a dwell time reported on Table 4, and then the sample was subjected to compression cycles with a maximum amplitude of *λ_max_* = 0.94 at different loading rates. The displacement pattern corresponding to Test 1 is shown in Figure 23a as an example. Only the positive part of the triangular wave was applied, and the strain rate considered to reach *λ* = 0.97 and during the triangular waves (i.e., from *λ* = 0.97 to *λ* = 0.94) are also reported in Table 4. It is noteworthy that the values of *λ* = 0.97 to *λ* = 0.94 can be assumed to be representative of the compression levels a bridge bearing would be subjected to under the effect of the permanent load, and the permanent load plus the loading of a heavy vehicle passing over the bridge.

Figure 23b shows the nominal stress–compression curve, which is highly nonlinear, although it is characterised by a low hysteresis. The response is slightly affected by the rate of deformation, with higher rates corresponding to a stiffer behaviour. Figure 24 shows the resistivity against the compression ratio for Test 3 and Test 4 as an example.

An interpolating function is fitted to the observed compression Test 3 data with the aim of establishing a model relating Δ*ρ*/*ρ_0_*, being *ρ_0_* = 3.8 Ωm, to the compressive strain *ε* = *λ*−1. For this purpose, two functions are used, one linear function with *GF* = 2.1 valid up to *ε* = −0.04, and one quadratic function valid beyond this value. Similar results are obtained with Test 4 data. The expressions of the functions are shown in Figure 25. The standard deviation of these data point from the linear model is $\sigma_{\varepsilon }\;$= 0.005. A third test has been performed on the same specimen by imposing the random input shown in Figure 26a.

The obtained histories of stress and electrical resistance are shown in Figure 26b. It can be observed that the resistivity increases when the compression increases, and decreases when the compression decreases, showing again the reversible behaviour due to the Printex rubber compound. The relationship obtained from the previous tests to relate the changes of resistivity to the changes of compression (Figure 25) can be used to check whether it is possible to infer the changes of strain in the specimen by measuring only the changes of electrical resistivity.

Figure 26c shows the measured changes of resistivity vs. the imposed changes of strain. In the same figure, the model of Figure 25 is also plotted. Figure 26d shows the time history of the experimental and predicted compressive strains. It can be observed that the model predictions are in reasonable accord with the measured values of the resistivity, thus confirming that the Printex−based rubber compound is suitable for developing smart rubber bearings.

### 5.3. Compression Tests on Rubber Filled with Printex 12 Phr (Specimen with l_0_/d_0_= 0.17)

In order to investigate the influence of the aspect ratio of the rubber layer on the piezoresistive response of the system, a cylindrical specimen with a more slender rubber layer (*l*_0_ = 8.47 mm), compared to the one considered previously (*l*_0_ = 5 mm), was manufactured and tested. A measure often employed to characterise the geometry of rubber layers is the shape factor (SF), which is the ratio of the loaded surface area of the rubber layer to the total area free to bulge [6]. The SF values for the two specimens are, respectively, 2.5 (*l*_0_/*d*_0_ = 0.1) and 1.48 (*l*_0_/*d*_0_ = 0.17). It is also noteworthy that in the case of bonded discs of rubber, the compressive load induces significant shear strains. In particular, the shear strains are zero in the centre and increase radially with the maximum value being at the edges. Both the maximum and the averaged shear stresses increase with the SF for a given compression level. Thus, higher shear stresses are expected in the specimen with a higher shape factor.

The specimen with SF = 1.48 was compressed by 0.5 mm steps (Δ*λ* = 0.058) up to the desired compression ratio of *λ* = 0.64. A dwell time of 60 s was considered between each further application of compression. Three identical cycles were imposed. Figure 27 shows the plot of the nominal compressive stress vs. the compression ratio obtained for all cycles. In this case, the stress−softening is negligible.

Figure 28 shows the electrical resistivity as a function of the compression ratio during each of the 3 cycles. Additionally, in this case, it can be observed that there is a highly nonlinear behaviour, especially for values of the compression ratio lower than *λ* = 0.8, whereas small changes of resistivity can be observed for higher values.

Another test was performed on the same sample. It was compressed up to *λ* = 0.97, then the compression was kept constant for a dwell time reported on Table 5, and then the sample was subjected to cycles of deformation with a maximum amplitude of *λ* = 0.94 for the different deformation rates as shown in Table 5. The displacement pattern applied in Test 1 is shown in Figure 29a. Figure 29b shows the stress–compression ratio curves, whereas Figure 30 shows the resistivity vs. the compression ratio obtained for Tests 3 and 4.

Figure 31 shows the normalised variation of the resistivity over the initial value at zero strain *ρ*_0_ = 6.6 Ωm vs. the compression strain *ε = λ−1*. The observed data are interpolated using a quadratic function. Figure 31 shows the data and the model fitted to the data. In order to provide a measure of the sensing sensitivity and precision, a linear model has also been fitted to the data in the range between *λ* = 1 and *λ* = 0.97, resulting in *GF* = 11.53 and $\sigma_{\varepsilon }\;$= 0.0085.

A third test was performed on the same specimen by imposing the input of Figure 32a. Figure 32b shows the stress and electrical resistance behaviour due to the applied displacement pattern. It can be observed that the resistivity increases when the compression increases and decreases when the compression decreases, showing again the reversible behaviour of the compound. Figure 32b shows that the strain values predicted using the resistivity measurements and the model of Figure 31 are in good agreement with the measured ones, thus confirming that this compound has a good potential to be used to develop smart rubber bearings.

## 6. Summary and Discussion of Results

Figure 33 compares the results of the experimental uniaxial tests performed on the specimens made with carbon black− and Printex 15 phr−filled compounds. The resistivity variation with the change of the extension ratio during the loading and the unloading path of the uniaxial test confirms that the Printex−filled compounds have a strong potential for sensing applications, thanks to their reversible behaviour.

The results presented in Section 3 and Section 4 also show that simple linear models can be used to relate the change of resistivity to the changes of strain in the material in the small strain range, i.e., up to *λ* = 2 for uniaxial tests, *γ* = 0.7 for DBS tests, and *λ* = 0.97 for the compression test. Table 6 summarizes the gauge factors and the standard deviation in terms of *ε* for the tensile specimen made with Printex 15 phr and the two compressive specimens made with Printex 12 phr. The sensitivity in compression is significantly lower than in tension. Moreover, the specimen with a higher aspect ratio is characterised by a higher sensitivity of the resistivity to the compression strains. This result can also be visualised by looking at Figure 34. A possible explanation of this behaviour can be provided by noting that for a given value of compression, the bulging of a rubber layer is more significant for the case of low shape factors, and that the bulging increases the length of the conductive path (see Figure 35). However, other factors may also affect the changes of resistivity, such as the fact that in a bonded rubber layer, the compression causes at local level significant shear strains, which are dependent on the shape factor of the layer.

## 7. Conclusions and Future Studies

This study has investigated the piezoresistive properties of three different compounds, one incorporating carbon black and the others with Printex XE2 as the filler. Specimens of these compounds have been electromechanically characterized under different loading conditions, namely tensile, compressive, and shear. The main results of the study are:—only the compounds filled with Printex XE2 have a significant potential for being used to develop smart rubber bearings, thanks to their reversible behaviour;—simplified models relating the change of resistivity to the changes of strain have been successfully used to infer the state of strain in the rubber under random loading scenarios based on electrical resistivity measurements;—the aspect ratio of the rubber layer significantly affects the piezoresistive behaviour under compressive loading. Finite element analyses (FEA) using a coupled electrical–mechanical model of the specimen should be carried out to confirm this.

Future studies will investigate further the development of smart rubber bearings and the eventual fatigue failure of elastomeric component under cyclic loads. In particular, alternative rubber compounds will be developed, using other volume fractions of Printex, or using alternative fillers, such as carbon nanotubes or graphene, in order to reach a more optimal behaviour in terms of mechanical and piezoresistive properties. For example, further research is needed to increase the gauge factor under small strains, so that the rubber exhibits more significant changes of resistivity for small loadings. This would unlock the potential to also detect the effect of loadings of lower intensity, not only exceptional loadings or heavy vehicle loadings.

Future experiment tests should include the effect of change in the temperature on the resistivity, assuming the temperature range of interest relevant to the service condition, and more thorough investigations on rate effects and on repeatability and statistical variability of the experimental response. The final material will also be characterised looking at creep rate, fatigue life, and strength properties. Once the optimal rubber compound and the optimal shape factor values is identified, two scaled laminated bearings will be manufactured and used to develop a scaled prototype of a simply supported bridge deck resting on the smart bearings. This will be useful to demonstrate the capabilities of the proposed technology and its use in real applications for bridge monitoring and as weigh in motion system.

## Figures and Tables

**Figure 1 sensors-23-03150-f001:**
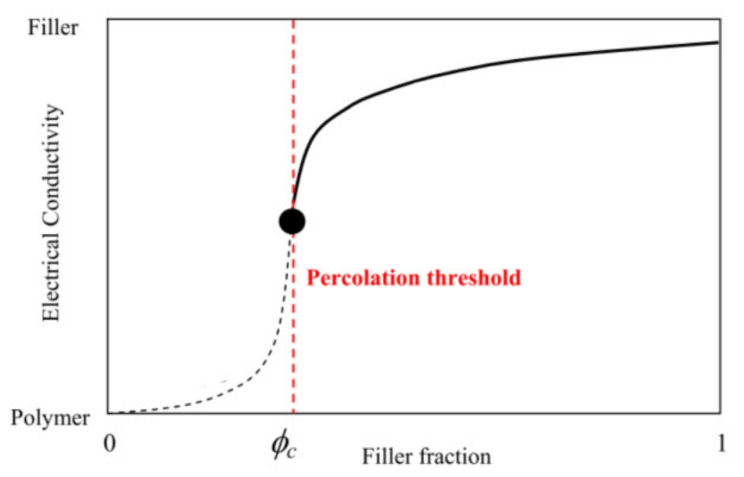
Electrical conductivity as a function of filler fraction.

**Figure 2 sensors-23-03150-f002:**
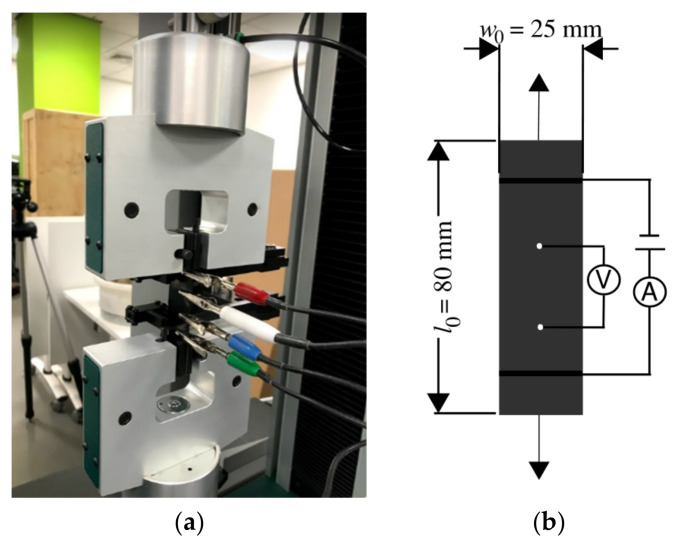
(**a**) Tensile test and resistance measurement, (**b**) Scheme of electrical resistance interrogation system.

**Figure 3 sensors-23-03150-f003:**
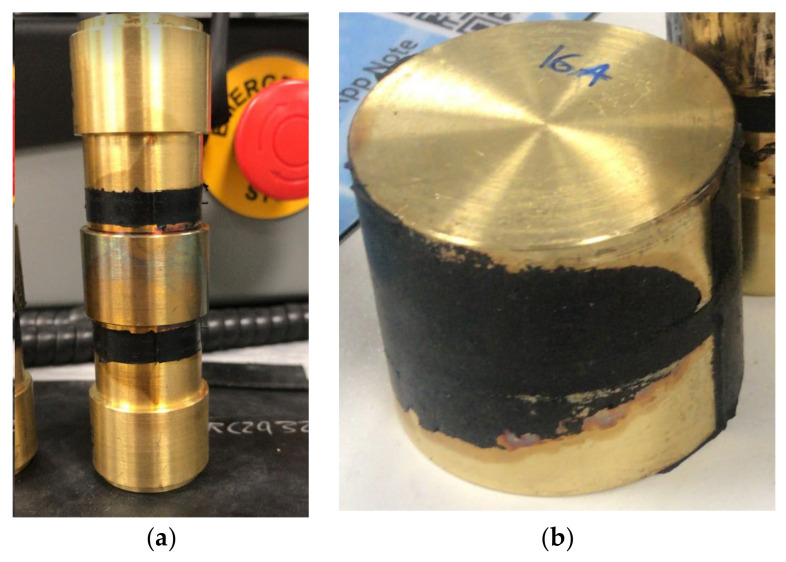
(**a**) DBS specimen, (**b**) Compressive specimen.

**Figure 4 sensors-23-03150-f004:**
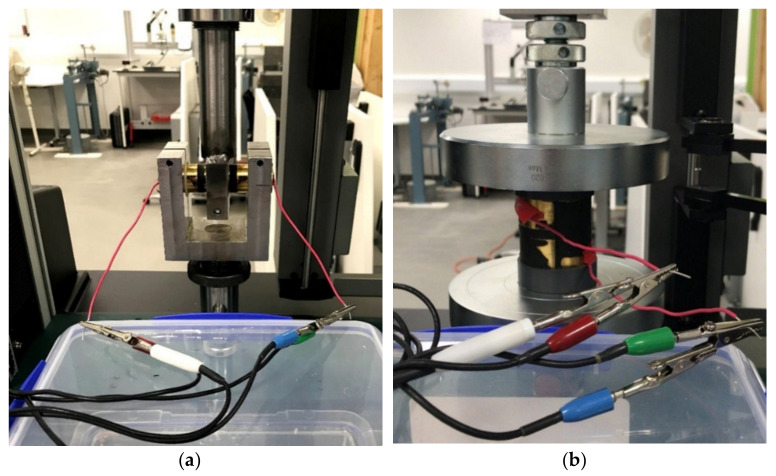
Resistance measurement, (**a**) DBS test equipment, (**b**) Compressive test equipment.

**Figure 5 sensors-23-03150-f005:**
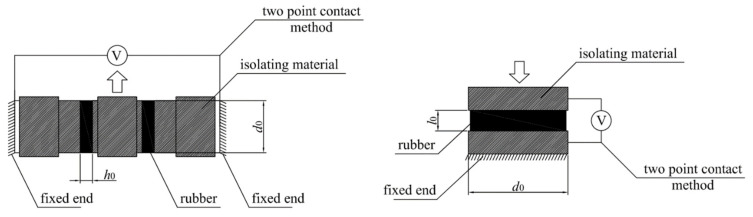
A schematic illustrating the two−point probe electrical resistance measurement technique for DBS and compressive specimens.

**Figure 6 sensors-23-03150-f006:**
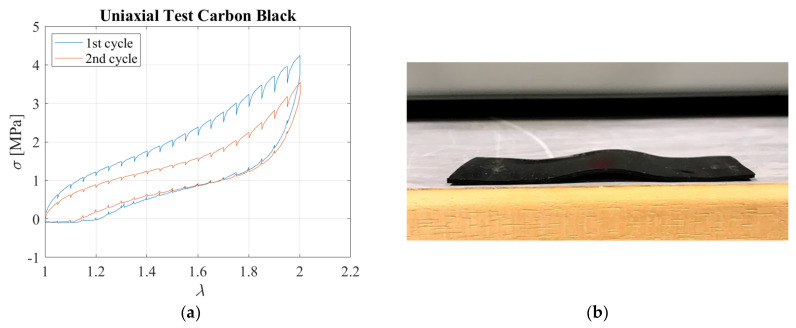
(**a**) Tensile stress−extension curves for two cycles; (**b**) Sample immediately after tensile test.

**Figure 7 sensors-23-03150-f007:**
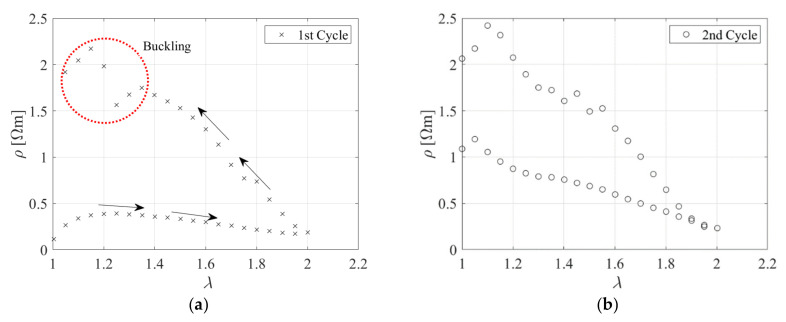
Resistivity as a function of the extension ratio during the (**a**) 1^st^ cycle and (**b**) 2nd cycle.

**Figure 8 sensors-23-03150-f008:**
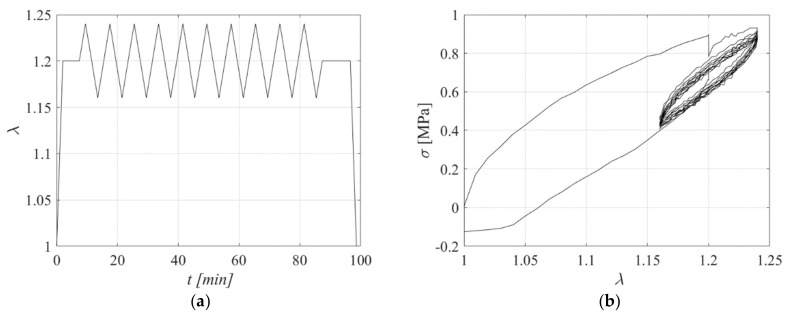
(**a**) Displacement pattern, (**b**) Stress−extension curve.

**Figure 9 sensors-23-03150-f009:**
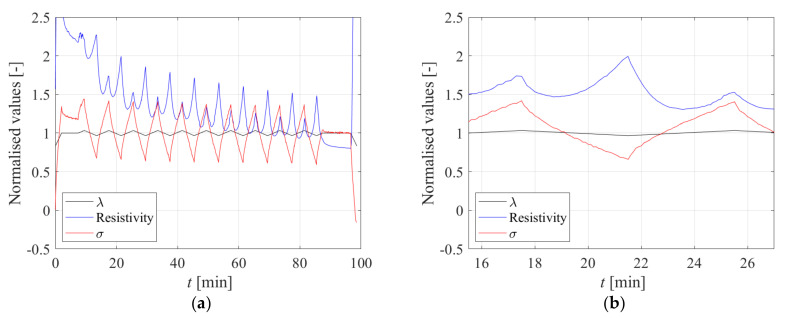
Resistivity and stress response due to the applied displacement pattern; (**a**) Full time history response, (**b**) Single cycle time history response.

**Figure 10 sensors-23-03150-f010:**
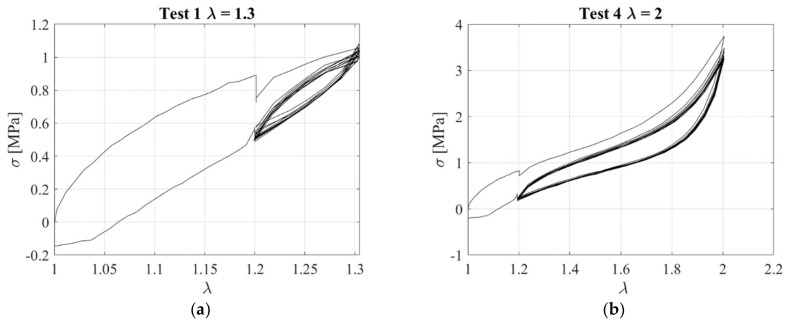
Stress−extension curve for Test 1 (**a**) and Test 4 (**b**).

**Figure 11 sensors-23-03150-f011:**
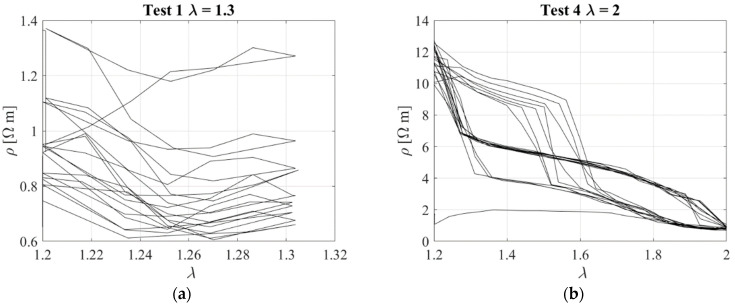
Resistivity vs. stretch ratio Test 1 (**a**) and Test 4 (**b**).

**Figure 12 sensors-23-03150-f012:**
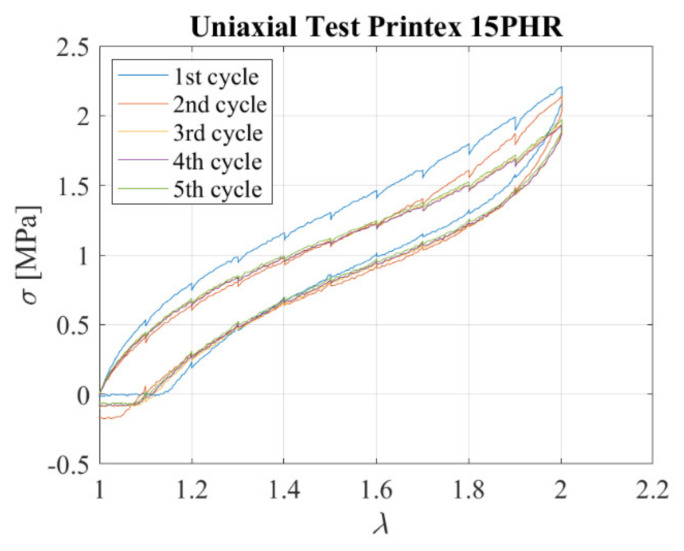
Tensile stress-extension curves for 5 cycles: Nominal stress vs extension.

**Figure 13 sensors-23-03150-f013:**
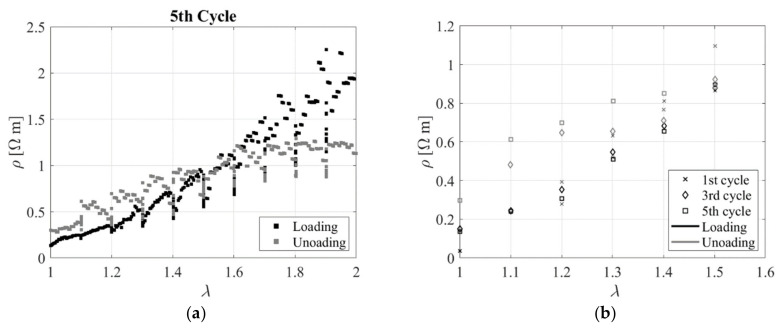
(**a**) Resistivity vs. extension ratio during 5th cycle, and (**b**) Resistivity vs. extension ratio during the 1st, 3rd and 5th loading cycles.

**Figure 14 sensors-23-03150-f014:**
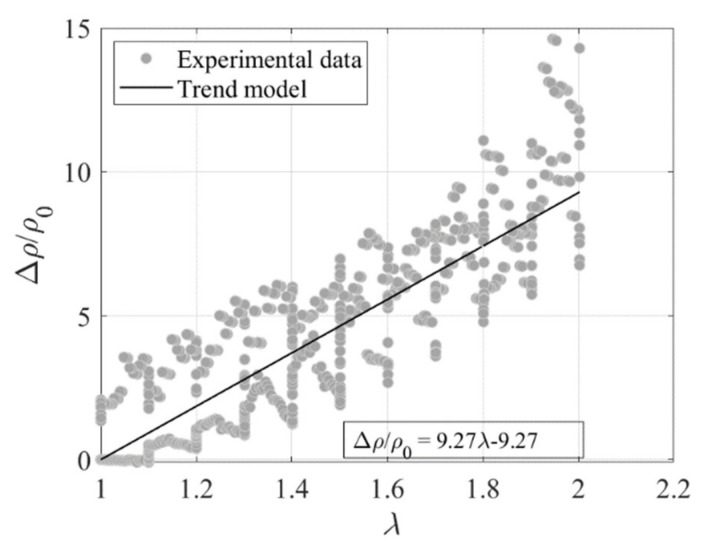
Variation of normalised resistivity with stretch ratio.

**Figure 15 sensors-23-03150-f015:**
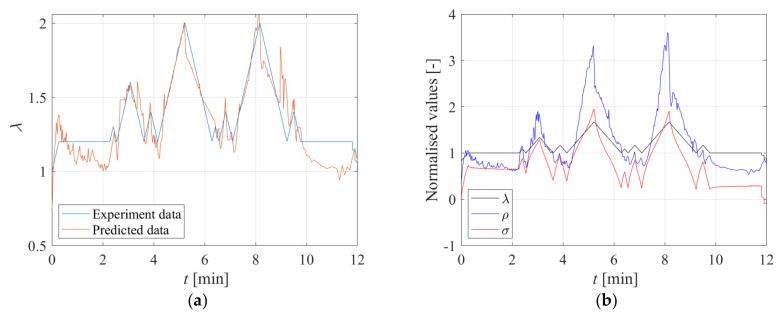
(**a**) Extension ratio vs. time: comparison between experiment and predicted data; (**b**) Time history of normalised displacement, stress, and electrical resistance.

**Figure 16 sensors-23-03150-f016:**
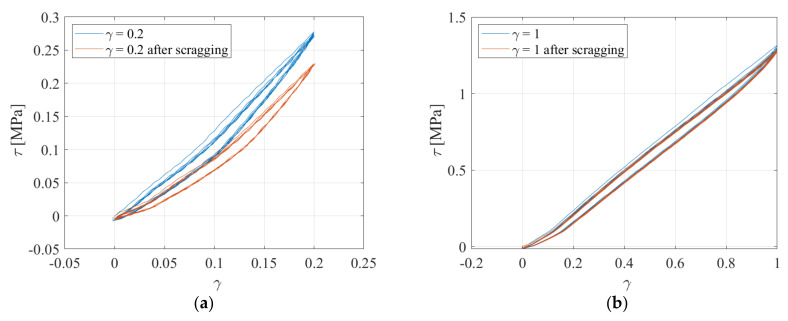
Stress−strain curves for (**a**) 20% and (**b**) 100% of nominal shear strain.

**Figure 17 sensors-23-03150-f017:**
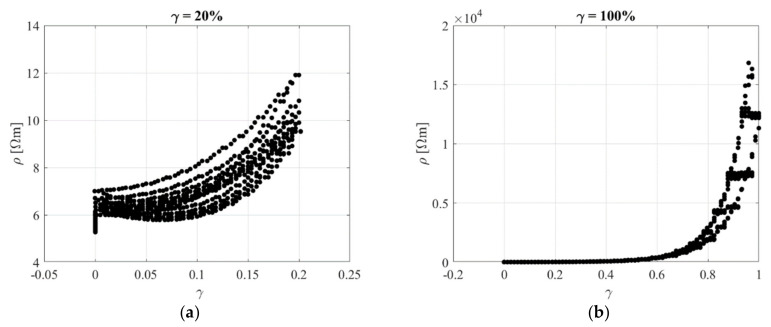
Resistivity vs. shear strain for different tests (**a**) *γ_max_* = 0.2 and (**b**) *γ_max_* = 1.

**Figure 18 sensors-23-03150-f018:**
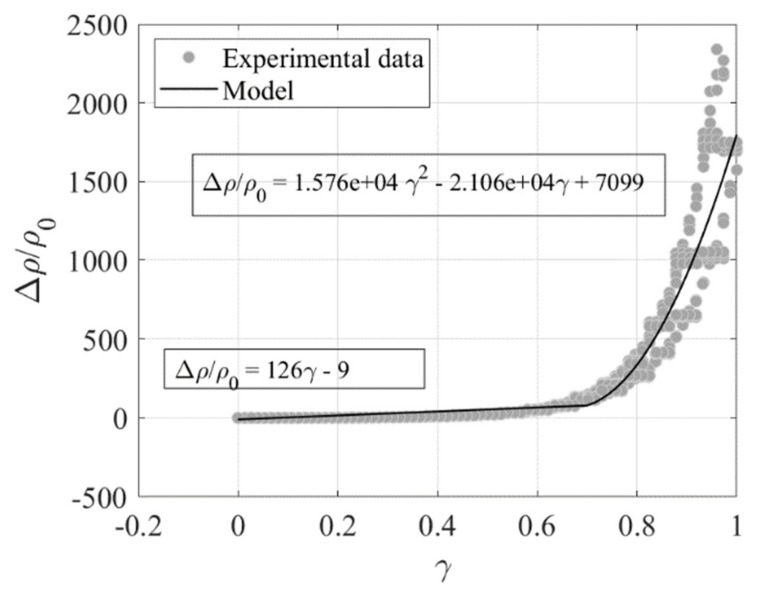
Variation of resistivity vs. shear strain.

**Figure 19 sensors-23-03150-f019:**
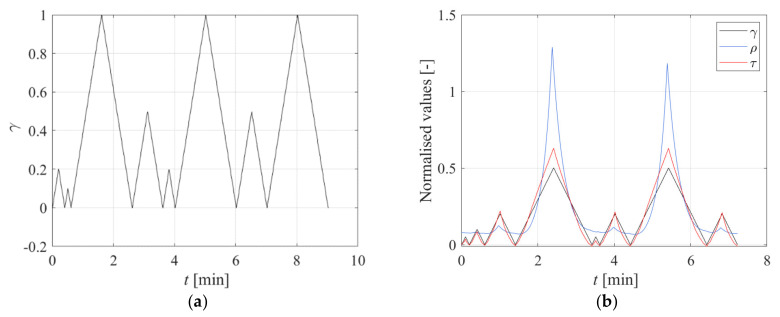
(**a**) History of input, (**b**) History of normalised input, stress, and resistivity.

**Figure 20 sensors-23-03150-f020:**
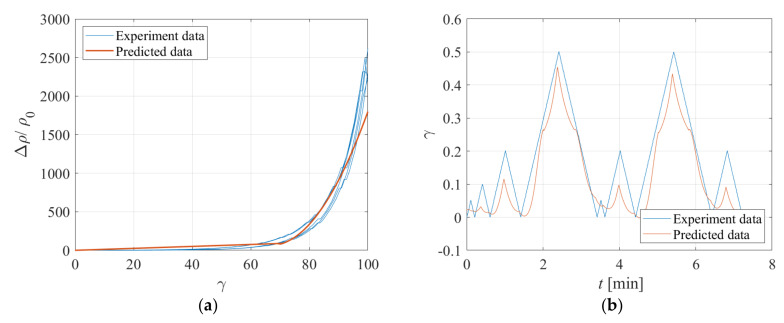
Comparison between experiment and predicted data: (**a**) Variation of resistance vs. strain, (**b**) Strain vs. time.

**Figure 21 sensors-23-03150-f021:**
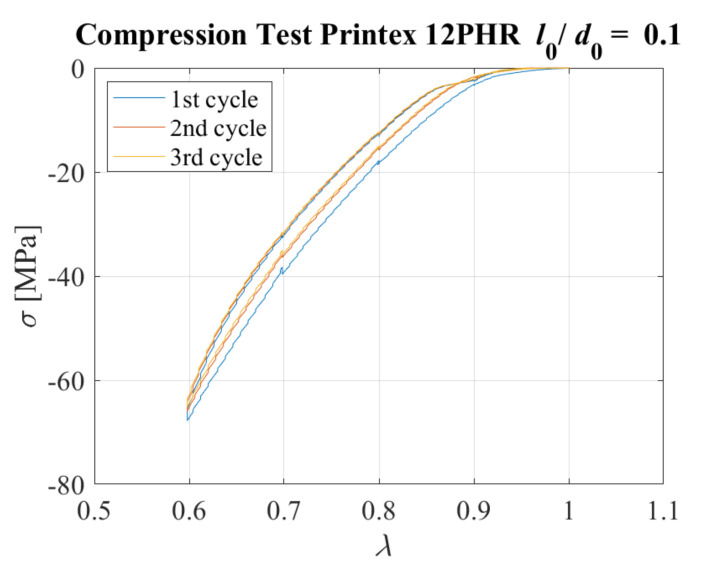
Compressive stress−strain curves for 3 cycles. Specimen with *l*_0_/*d*_0_ = 0.1.

**Figure 22 sensors-23-03150-f022:**
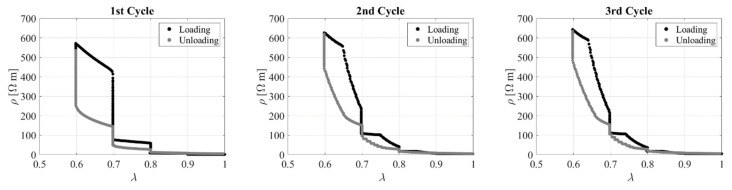
Resistivity vs. compression ratio during 3 cycles. Specimen with *l*_0_/*d*_0_ = 0.1.

**Figure 23 sensors-23-03150-f023:**
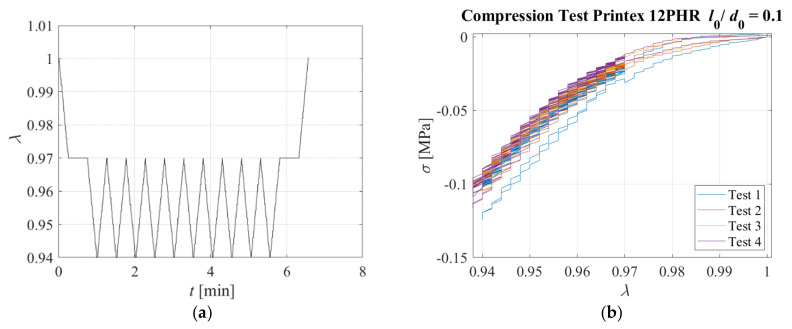
(**a**) Displacement pattern, (**b**) Stress−compression ratio relations for each triangular test. Specimen with *l*_0_/*d*_0_ = 0.1.

**Figure 24 sensors-23-03150-f024:**
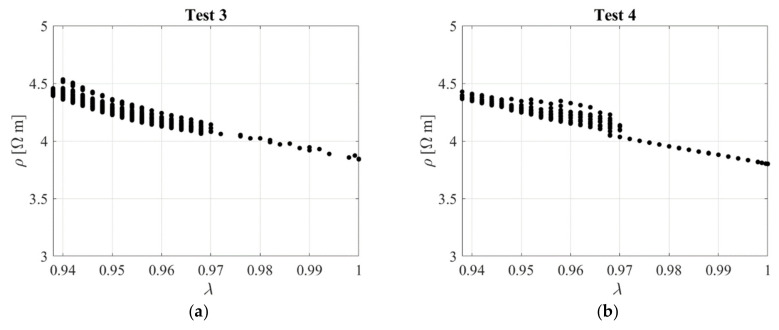
Resistivity against compression ratio: (**a**) Test 3 and (**b**) Test 4. Specimen with *l*_0_/*d*_0_ = 0.1.

**Figure 25 sensors-23-03150-f025:**
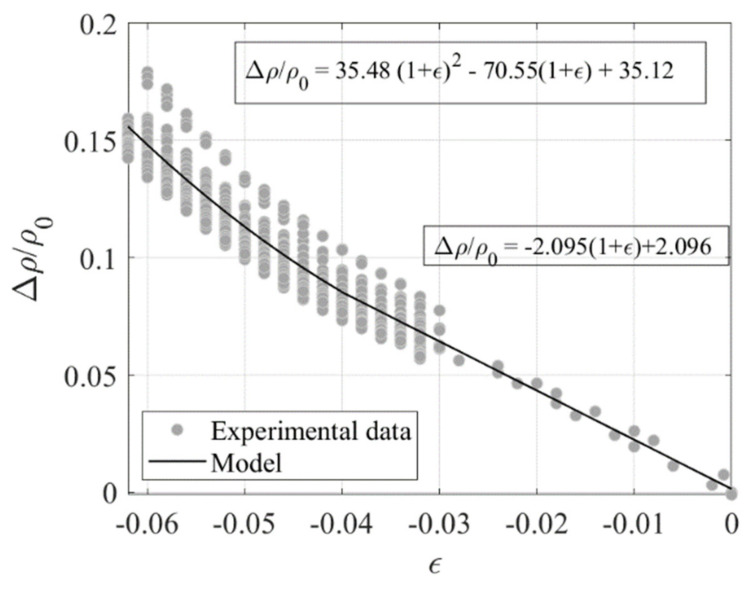
Variation of resistivity ratio vs. compression strain.

**Figure 26 sensors-23-03150-f026:**
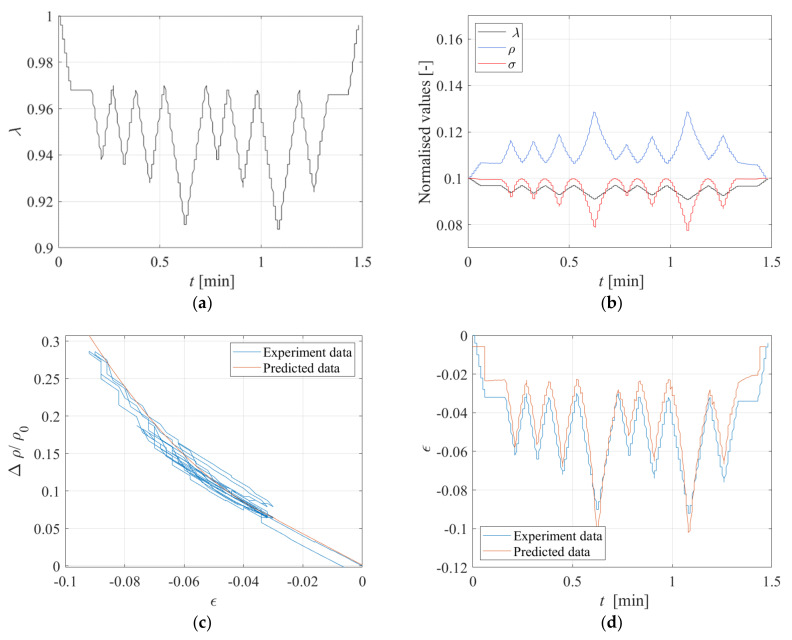
(**a**) Input random, (**b**) Time history of normalised displacement, stress, and electrical resistance under the displacement input of Figure 26a. Comparison between experiment and predicted data: (**c**) Variation of resistivity vs. strain, and (**d**) Strain vs. time.

**Figure 27 sensors-23-03150-f027:**
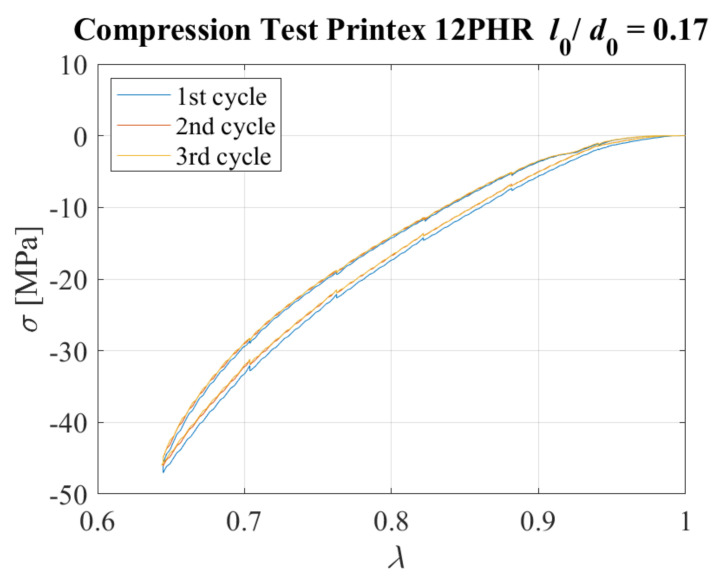
Compressive stress−strain curves for 3 cycles. Specimen with *l*_0_/*d*_0_ = 0.17.

**Figure 28 sensors-23-03150-f028:**
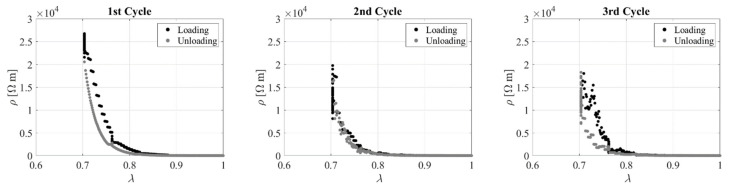
Resistivity vs. compression ratio during 3 cycles. Specimen with *l*_0_/*d*_0_ = 0.17.

**Figure 29 sensors-23-03150-f029:**
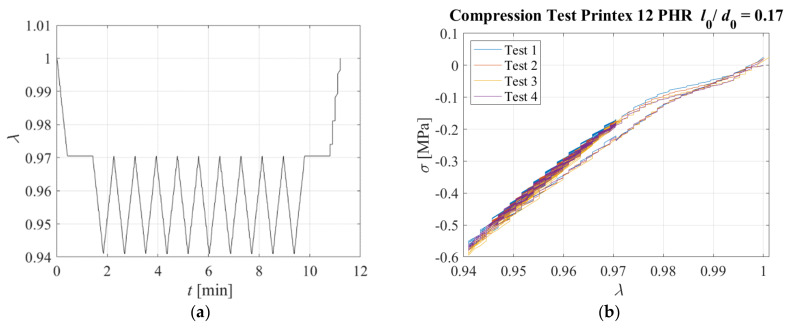
(**a**) Displacement pattern, (**b**) Stress−compression ratio relations for each triangular test. Specimen with *l*_0_/*d*_0_ = 0.17.

**Figure 30 sensors-23-03150-f030:**
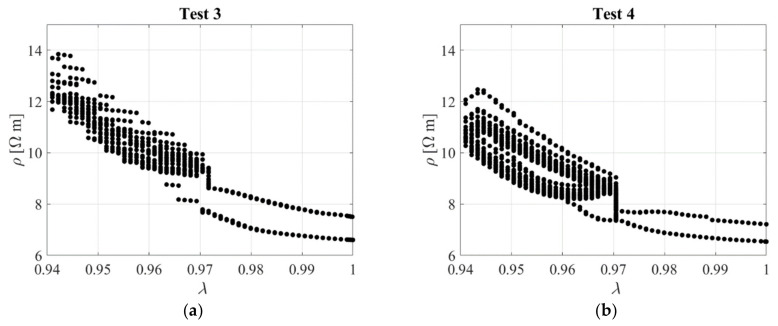
Resistivity against compression ratio: (**a**) Test 3 and (**b**) Test 4. Specimen with *l*_0_/*d*_0_ = 0.17.

**Figure 31 sensors-23-03150-f031:**
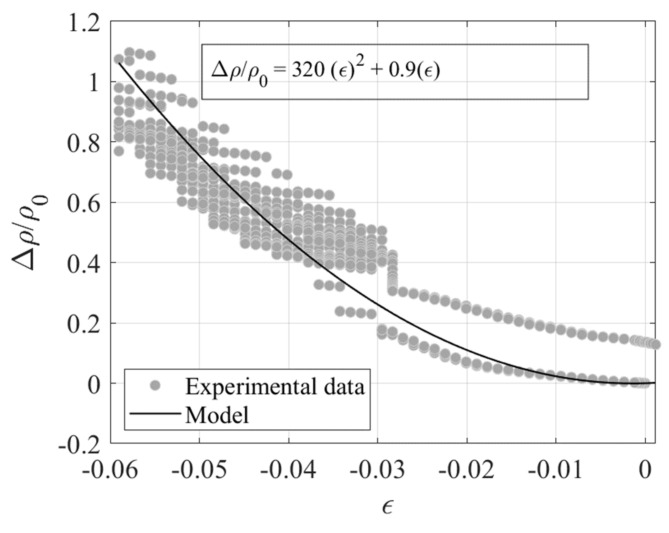
Variation of normalised resistivity vs compressive strain.

**Figure 32 sensors-23-03150-f032:**
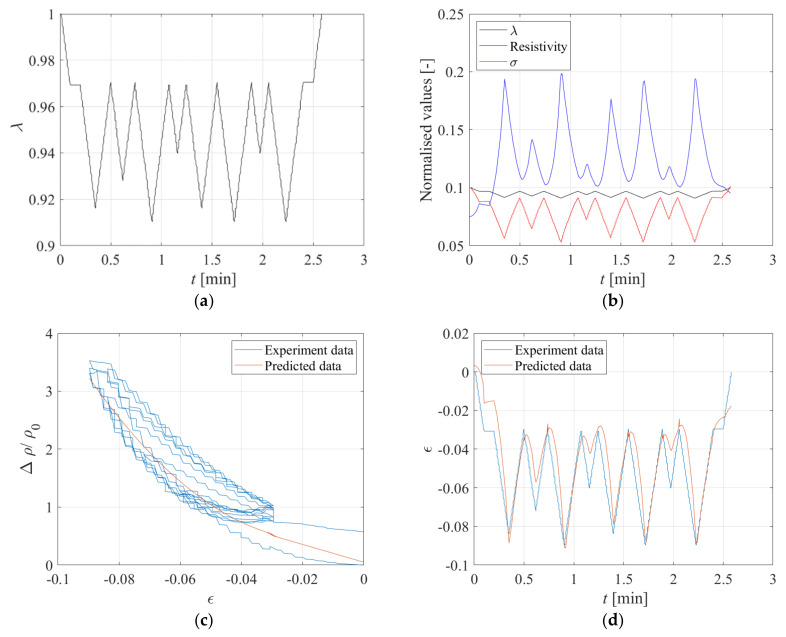
(**a**) Input random, (**b**) Time history of normalised displacement, stress, and electrical resistance under the displacement input of Figure 32a. Comparison between experiment and predicted data: (**c**) Variation of resistivity vs. strain, and (**d**) Strain vs. time.

**Figure 33 sensors-23-03150-f033:**
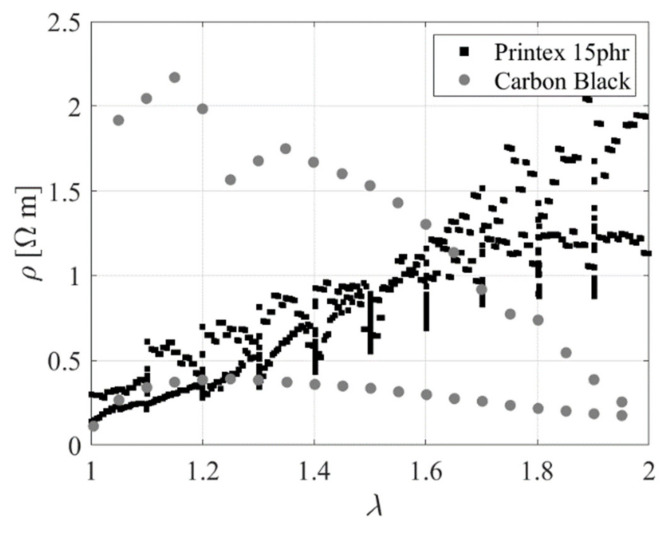
Resistivity as a function of the extension ratio for carbon black− and Printex 15 phr−filled rubber.

**Figure 34 sensors-23-03150-f034:**
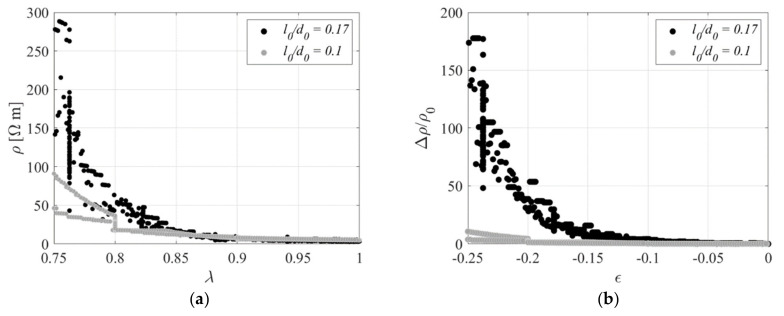
Effect of the shape factor: (**a**) Resistivity vs. compressive strain, (**b**) Normalised resistivity change vs. compressive strain.

**Figure 35 sensors-23-03150-f035:**
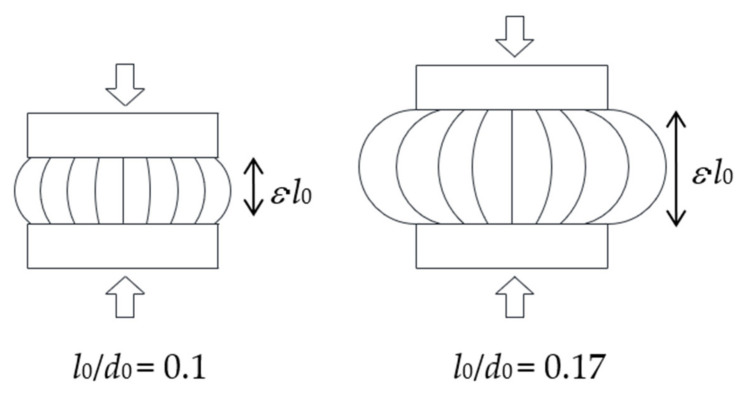
Effect of bulging on deformed geometry of the compressive specimens.

**Table 1 sensors-23-03150-t001:** Formulation of the filled rubber compound in weight (parts per hundred rubber).

Material	Abbreviation		
Standard Malaysian Rubber	SMR		
High abrasion furnace	HAF		
Hexyl phenyl phenylenediamine	HPPD		
Cyclohezyl benzothiazxy sulphenamide	CBS		
Tertiary butyl benzothiazole sulfenamide	TBBS		
**Ingredients**	**Parts per hundred of rubber (phr)**		
	**carbon black 70 phr**	**Printex XE2** **15 phr**	**Printex XE2** **12 phr**
NR (SMR CV60)	100	100	100
Carbon black (N330 HAF)	70	-	-
Printex XE2	-	15	12
Stearic acid	2	2	2
Zinc oxide	10	7	7
Antioxidant (HPPD)	1	-	-
Cobalt naphthenate	3	-	-
Accelerator (CBS)	0.8	-	-
6PPD	-	1.5	1.5
Antilux 654	-	1.5	1.5
Manobond 740 C	-	0.75	0.75
TBBS	-	1.5	1.5
Sulphur	4	1.5	1.5

**Table 2 sensors-23-03150-t002:** Test pieces and loading conditions.

Test Pieces	Loading				
	Tensile			Shear	Compression
	Full Cycle	Triangular	Random		
Carbon black tensile specimen	✓	✓			
Printex 15 phr tensile specimen	✓		✓		
Printex 12 phr Double Bonded Shear (DBS) specimen				✓	
Printex 12 phr compressive disc specimen *l*_0_/*d*_0_ = 0.1					✓
Printex 12 phr compressive disc specimen *l*_0_/*d*_0_ = 0.17					✓

**Table 3 sensors-23-03150-t003:** Axial elongation and velocity values performed during the tests.

	Test 1	Test 2	Test 3	Test 4
***λ* [-]**	1.3	1.4	1.6	2
$\dot{\lambda \;}$ **[s^−1^]**	0.05	0.1	0.05	0.05

**Table 4 sensors-23-03150-t004:** Velocity values and dwell time applied during the tests.

	Test 1	Test 2	Test 3	Test 4
${\dot{\lambda }}_{1stramp}\;$ **[s^−1^]**	0.0017	0.0017	0.0075	0.0075
${\dot{\lambda }}_{2ndramp}\;$ **[s^−1^]**	0.0017	0.0017	0.0017	0.01
**Dwell time [s]**	30	6	6	3

**Table 5 sensors-23-03150-t005:** Velocity values and dwell time applied during the tests.

	Test 1	Test 2	Test 3	Test 4
${\dot{\lambda }}_{1stramp}\;$ **[s^−1^]**	0.0017	0.0017	0.0075	0.0075
${\dot{\lambda }}_{2ndramp}\;$ **[s^−1^]**	0.0017	0.0017	0.0017	0.01
**Dwell time [s]**	30	6	6	3

**Table 6 sensors-23-03150-t006:** Gauge factor and standard deviation of linear models adopted.

Specimens	*GF*	σ_ε_
Printex 15 phr tensile specimen	9.27	0.16
Printex 12 phr compressive specimen *l*_0_/*d*_0_ = 0.1 (SF = 2.5)	2.10	0.005
Printex 12 phr compressive specimen *l*_0_/*d*_0_ = 0.17 (SF = 5.88)	11.5	0.0085

## Data Availability

The data presented in this study are available on request from the corresponding author.

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
