# Peer review of "Self-Sensing Rubber for Bridge Bearing Monitoring"

_sensors, 2023, doi:10.3390/s23063150_

Round 1

Reviewer 1 Report

Elastomeric bearing is of paramount importance to bridge safety assessment and management. This manuscript attempts to explore the feasibility of exploiting the piezo-resistive properties of filled rubber to monitor the condition of elastomeric bearings. Various filled rubber compounds tested were conducted to evaluate their piezo-resistive properties. This work is of great benefit for assessing the bridge responses to external loadings. Some comments are as follows:

1. For rubber filled with Printex 15 Phr, the uniaxial tensile tests show that apart from the first cycle, all subsequent cycles are almost indistinguishable. What is the reason/mechanism of the difference? It is better to discuss more deeply.

2. For each test, only one sample with several loading/unloading cycle is performed. Thus, the test result and its corresponding fitting curve are lack of universality. It should be better to increase the number of test sample or use finite element method to validate the test result.

3. The last section (“5. Conclusion and future studies”) is too long, and it needs to be simplified for highlighting the main achievements and innovations of this research.

4. There is a little error in the section number.

Reviewer 2 Report

It is an original paper dealing with “Self-sensing rubber for bridge bearing monitoring “. It is quite well organized and its language is quite satisfactory. However, there are some minor and major comments below to help the readers to be more beneficial from the paper.

1.       Page 2, line 38, the author need to clarify for the first time the LVDT and then use the abbreviation throughout the paper. Similarly, in line 41, what is the abbreviation of the PVDF.

2.       As the authors used piezoresistive sensor for structural health monitoring, they should explain in the paper what is the piezoresistive sensors and what is the advantage of these sensors in comparison with other sensors like FPGA, , Acoustic emission. In this regards there are a lot of references that the author can refer to them. In line 53, the authors can write piezoresistive sensors comprises of conductive filler dispersed in an insulating polymer matrix which create a conductive network. During the damage extension or crack propagation the conductive network beaks and the electrical resistance changes.

(a)    Damage sensing of adhesively-bonded hybrid composite/steel joints using carbon nanotubes. Composites Science and Technology, 2011, 71(9), 1183-1189

(b)   Impedance analysis for condition monitoring of single lap CNT-epoxy adhesive joint. International Journal of Adhesion and Adhesives, 2019, 88, 59-65.

(c)     Integration of carbon nanotube sensing skins and carbon fiber composites for monitoring and structural repair of fatigue cracked metal structures. Composite Structures, 2018, 203, 182-192.

(d)   Structural health monitoring of adhesive joints under pure mode I loading using the electrical impedance measurement. Engineering Fracture Mechanics, 2021, 245, 107585

3.        It is not clear in the paper how the prizoresistive sensor has been prepared. How the conductive materials added to the elastomer. In order to have a have proper and precise percolation threshold the conductive materials must be added to the polymer matrix appropriately. It means that the conductive materials should be dispersed or distributed though out the polymer matrix homogeneously. In this paper there is no SEM picture which shows how the conductive materials distributed in the matrix. There is no illustration of electro-mechanical measurement set-up.

4.       Why the authors used 4 probe method instead of 2 probe method to see the change in electrical resistance?

5.       Are there any standards for DBS sample and compression sample preparation? Please clarify this by referring to appropriate references

6.       Use bullets to emphasize the main achievements of the paper

Round 2

Reviewer 2 Report

Thank you for referring to the comments, accordingly. The manuscript is now acceptable from my side without further review .